# Exploiting *In-Situ* Characterization for a Sabatier Reaction to Reveal Catalytic Details

**Simon Yunes \*, Urim Pearl Kim, Hoang Nguyen**  **and Jeffrey Kenvin**

Micromeritics Instrument Corporation, 4356 Communications Drive, Norcross, GA 30093, USA;
pearl.kwon@micromeritics.com (U.P.K.); hoang.nguyen@micromeritics.com (H.N.);
jeff.kenvin@micromeritics.com (J.K.)
\* Correspondence: simon.yunes@micromeritics.com

**Abstract:** *In situ* characterization of catalysts provides important information on the catalyst and the understanding of its activity and selectivity for a specific reaction. TPX techniques for catalyst characterization reveal the role of the support on the stabilization and dispersion of the active sites. However, these can be altered at high temperature since sintering of active species can occur as well as possible carbon deposition through the Bosch reaction, which hinders the active species and deactivates the catalyst. In situ characterization of the spent catalyst, however, may expose the causes for catalyst deactivation. For example, a simple TPO analysis on the spent catalyst may produce CO and $CO_2$ via a reaction with $O_2$ at high temperature and this is a strong indication that deactivation may be due to the deposition of carbon during the Sabatier reaction. Other TPX techniques such as TPR and pulse chemisorption are also valuable techniques when they are applied in situ to the fresh catalyst and then to the catalyst upon deactivation.

**Keywords:** *in situ*; TPX; sintering; catalyst; spent; chemisorption



## 1. Introduction

The *in-situ* Catalyst Characterization System (ICCS) [1,2] from Micromeritics and the Cirrus MKS [3] mass spectrometer prove to be a very powerful system when combined with the Micromeritics Flow Reactor (FR) [1]. The ICCS allows a complete characterization of the catalyst *in situ*. For example, a necessary technique often used is temperature programmed reactions (TPX) [4]. This technique reveals important properties of the catalyst, such as the effects of the support material in stabilizing the active species to minimize sintering, a quantitative analysis to determine the quantity of the oxide used as active species, the active particle size that can easily be estimated by the mechanism of reduction shown by the TPR profile, etc. Another technique that should also be used for characterization is pulse chemisorption. This reveals the percentage of the active particles exposed for the reaction and thus predicts the activity and selectivity of the catalyst. For example, this can be implemented to reevaluate the catalyst, if deactivation or poisoning of the catalyst is suspected due to the reaction conditions.

A mass spectrometer connected to the exhaust of the flow reactor was utilized for online detection and quantification of the reaction products. The Cirrus II MKS mass spectrometer is a reliable system capable of detecting up to 200 amu and was equipped with a heated capillary that provides temperature control up to a maximum of 150 °C. The temperature controlled capillary prevents unwanted condensation vapor products and allows for the accurate sampling of the product mixture. The combination of these three instruments creates a powerful toolset for researchers in catalysis and provides catalyst characterization and evaluation all on one single system (Figure 1).

$$CO_2 + 2H_2 - - - - - -C + 2H_2O.$$

$$2CO_2 + 5H_2 - - - - - -CH_4 + CO + 3H_2O$$

$$CO_2 + H_2 - - - - - -CO + H_2O$$

$$CH_4 + CO_2 - - - - - -2CO + 2H_2$$

$$CO + H_2 - - - - - -C + H_2O$$

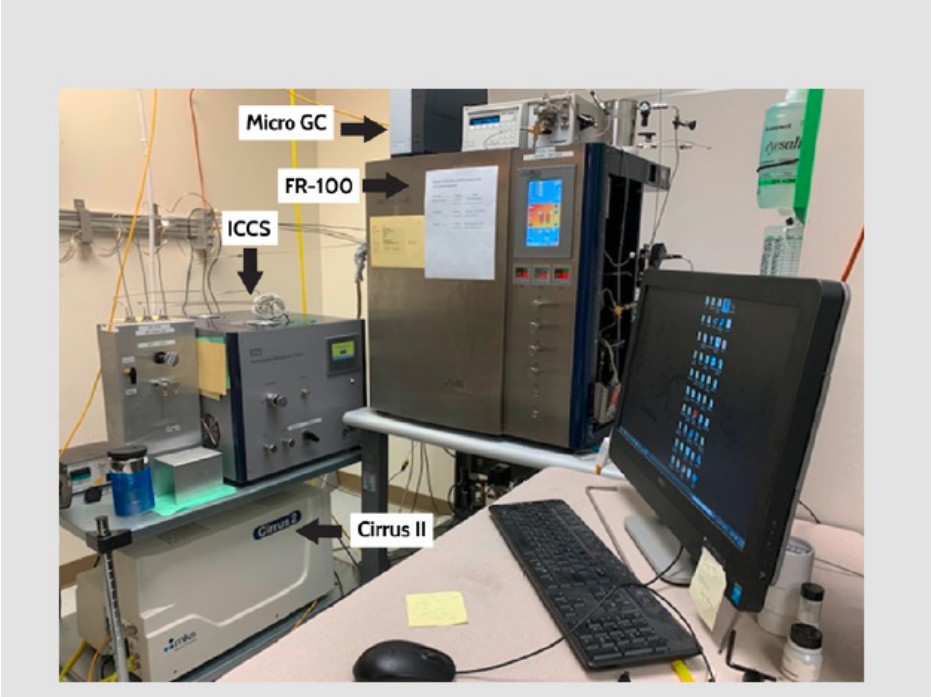

**Figure 1.** The set of 3 instruments connected together.

A typical Sabatier [5] reaction for the conversion of carbon dioxide to methane ($CO_2 + 4H_2 \rightleftharpoons CH_4 + 2H_2O$) was selected to demonstrate the capabilities for this combination of instruments. This reaction is widely used and well established, converting $CO_2$ under the effect of hydrogen to more useful products such as $CH_4$ and syngas ($CO + H_2$).

Based on the numerous publications [6–14] on this subject, it can be found that different activity and selectivity of the catalyst can be observed by varying the reaction conditions. Several possible reactions can take place when reducing $CO_2$ with hydrogen as a function of pressure and temperature: For example, the first reaction shown on the right is the formation of carbon (graphite) via the Bosch reaction [15]. The overall reaction is a two-step reaction with a fast reverse water gas shift reaction, which is then followed by a rate limiting step as shown below.

$$CO_2 + H_2 - - - - - -CO + H_2O$$

$$CO + H_2 - - - - - -C + H_2O$$

## 2. Experimental Procedures, Materials and Methods

Traditionally, nickel supported catalysts are used for methanation. Here, 1.5 g of 13% CuO alumina supported commercial catalyst from Sigma-Aldrich Batch # MKCM8623 was used for all of the above mentioned reactions. A fresh sample was used for each experiment in order to avoid the effect of sintering and carbon deposition that can take place at elevated temperature and thus reduce the activity of the catalysts in subsequent analysis.

The FR-100 PID-Micromeritics microreactor was used for these experiments and the hot box temperature was set to 120 °C to avoid condensation of reaction products. The

liquid–gas separator (L/G) was set at 4 °C to condense and trap the water produced during the reaction before it enters the mass spectrometer capillary system.

A mixture of 200 mL/min of hydrogen and 50 mL/min of $CO_2$ was used as a feed for the Sabatier reaction.

After loading the catalyst into the 9-mm SS reactor and prior to any reaction studies, the ICCS was used to perform a TPR for the quantification of CuO and to obtain the reduction profile. This also establishes the reduction mechanism of the oxide (Figure 2). The reduction profile produced a peak related to the consumption of hydrogen by the oxide, which was of 22 mL of $H_2$ and that corresponds to 13% by weight of the oxide on the catalyst. The temperature at the peak maximum for the reduction was observed at 200 °C.

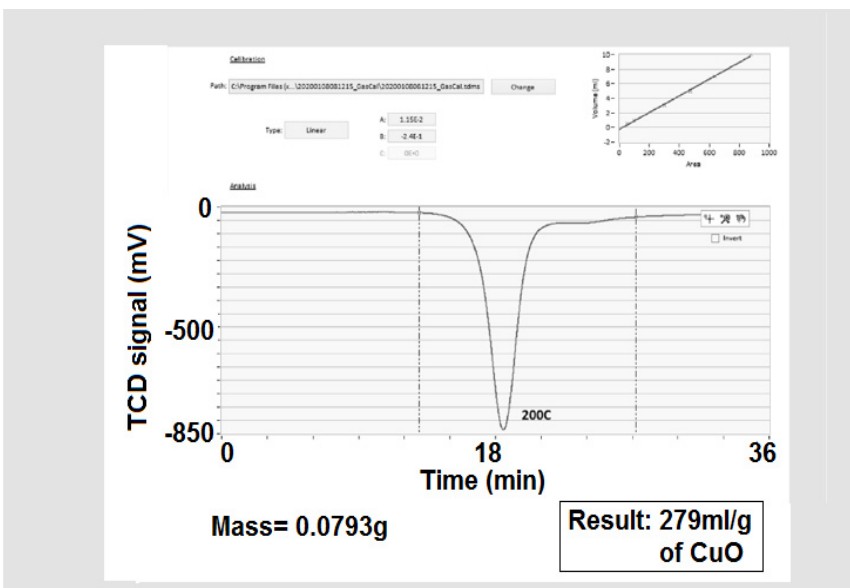

**Figure 2.** TPR profile on the 13% CuO catalyst.

The first in situ TPR shown in Figure 2 ensures that under these conditions the oxide was completely reduced to copper metal, which is one among the active elements for the Sabatier reaction. The TPR was performed under 10% $H_2$/Ar flow at 100 mL/min to 550 °C at 10 °C/min using the ICCS; TCD signal was collected.

The reaction conditions are unique to these experiments. The temperature was increased with a ramping rate of 2 °C/min from room temperature up to 600 °C. The reaction products were monitored online by the mass spectrometer as the temperature increased. The mass spectrometer was set to monitor the following mass per charge signals: 2, 28, 44, and 16, which correspond to $H_2$, CO, $CO_2$, and $CH_4$, respectively. The continuous monitoring of these signals allows users to visualize the reaction steps as a function of the temperature profile during the reaction.

This study was conducted to observe the effect of pressure on the reaction products of the reduction of $CO_2$. Four different experiments were carried out under the same gas mixture, sample size, and ramping temperature, but at different pressures.

The conversion was measured as the difference in intensities of $CO_2$ taken at the beginning and at the end of each experiment at 600 °C.

TPO analysis was conducted after the reaction of up to 600 °C using a flow of 100 mL/min of 10% Oxygen balance Helium. Mass per charge signals for $O_2$ (32), $CO_2$ (44), CO (28), and $H_2O$ (18) were collected. This test was performed to identify if carbon formation occurred at higher temperature. A TPR and a TPO analyses were also carried out on the fresh catalyst for comparison with the TPO profile done on the used catalyst. Differences between the two TPO profiles will reveal the presence of carbon deposition on the sample, if any.

## 3. Results

Reduction of $CO_2$ at atmospheric pressure:

Figure 3 corresponds to a mass spectrum at atmospheric pressure that includes signals for both reactants: $CO_2$ and $H_2$ as well as main products: CO and $CH_4$. It can be observed that, at atmospheric pressure, the reaction does not produce $CH_4$; only CO and water were produced. Conversion of $CO_2$ into products under the effect of $H_2$ produced the following intensities for both products (CO and $CH_4$). For CO, it was 192 (au), while it was zero for $CH_4$. There was no signal for water on the spectrum, as water was trapped by the liquid/gas separator (L/G).

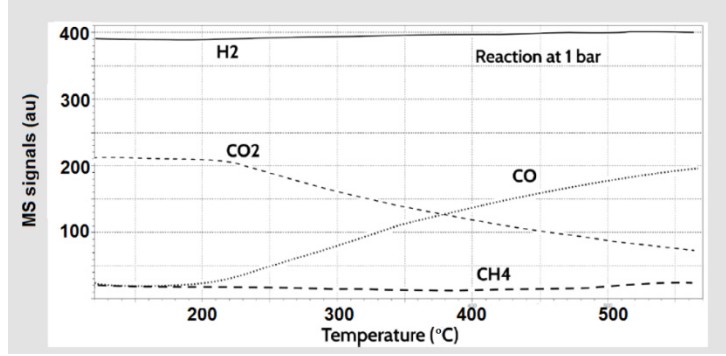

**Figure 3.** Spectrum of signals at 1 bar.

Water was produced during the reaction and separated from reaction products via condensation using the liquid/gas separator (L/G). The condensed phase was collected in a beaker and the quantity of the water could be used in the final mass balance for the reaction (Figure 4).

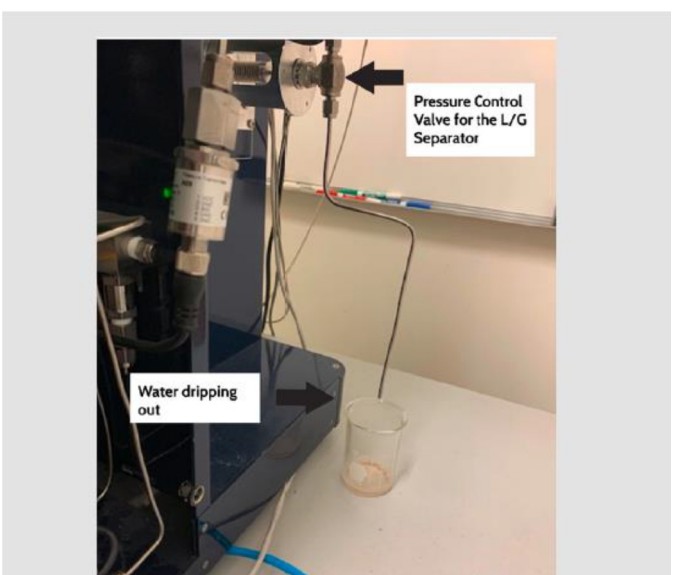

**Figure 4.** Water dripping out from the L/G separator.

Figure 5 represents signals for both reactants ($CO_2$ + $H_2$) at 10 bar of pressure. By the contrast at 10 bar, $CH_4$ was produced as shown on the spectrum by the signal of 16 amu. Water was trapped by the (L/G) separator for all experiments. This figure shows the production of $CH_4$ as well as CO. Conversion of $CO_2$ at 10 bar produced intensities for CO and $CH_4$ of 260 (au) and 210 (au), respectively. It can be concluded that the activity at 10 bar was somehow moderate to produce the main product ($CH_4$) and was of 1.3 times higher for CO at 600 °C.

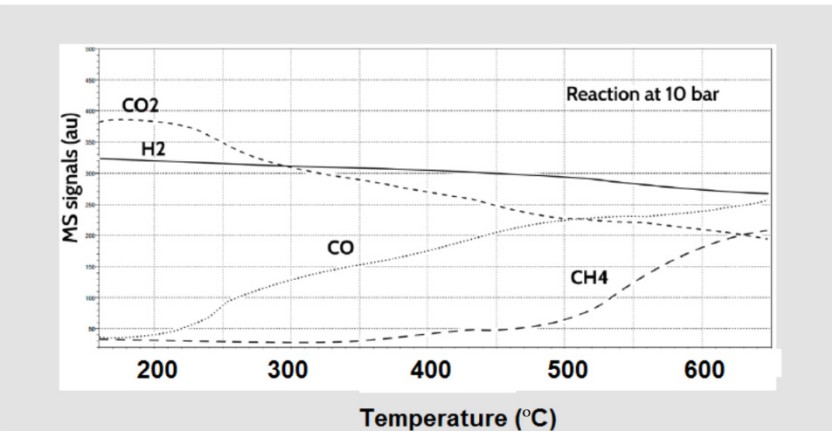

**Figure 5.** Spectrum of signals at 10 bar.

Figure 6 represents signals for the reaction at 20 bar. It can be observed from these results that as the pressure increases, the production of $CH_4$ increases as well. All studied reactions produced water and CO. This reaction produced intensities for CO and $CH_4$ of 310 (au) and 430 (au), respectively. The production of $CH_4$ was doubled as the pressure changed from 10 to 20 bar and was of 1.2 times higher for CO.

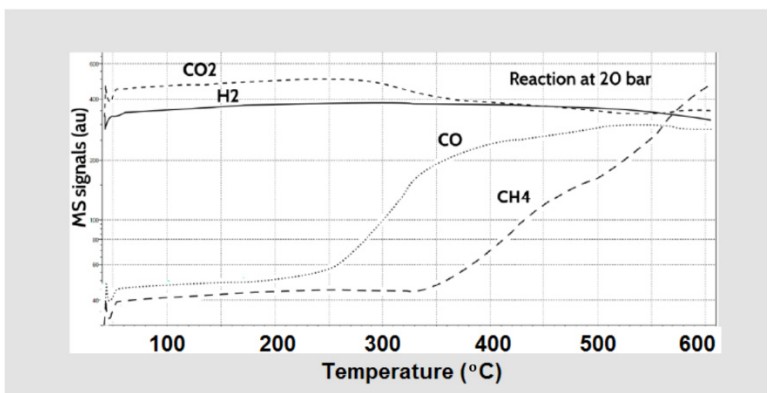

**Figure 6.** Spectrum of signals at 20 bar.

Figure 7 represents signals for the products at 30 bar. It was observed that increasing the pressure from 20 to 30 bar did not enhance the production of $CH_4$. Intensities for CO and $CH_4$ were of 170 (au) and 375 (au), respectively. However, it was observed that a larger amount of water was produced in this case, as it was seen dripping out from the L/G separator.

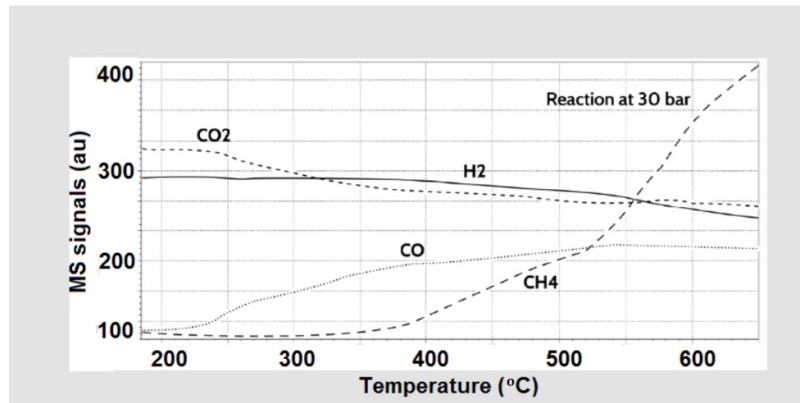

**Figure 7.** Spectrum of signals at 30 bar.

Figure 8: This figure illustrates the intensity of each product at 600 °C for these four reactions as a function of pressure increase from atmospheric pressure up to 30 bar.

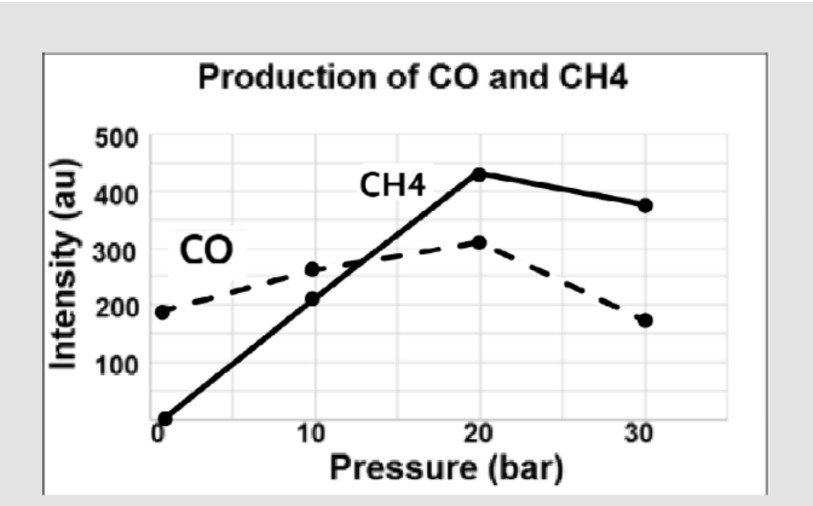

**Figure 8.** MS signals of the different elements of the reaction.

## 4. Results upon Deactivation, and Data Interpretation

Figure 9 shows a temperature programmed oxidation (TPO) profile on the spent catalyst. Comparisons with the TPO profile done on the fresh material after reduction (Figure 12) would indicate if there was any carbon formation from $CO_2$ at high temperature. Carbon deposition on the surface of the catalyst could block access to the active sites for the reaction. Oxidation of carbon during TPO at high temperature produces CO and/or $CO_2$. The TPO profile shows several peaks (Figure 9) up to 300 °C, and some of them were not identified by the mass spectrum. CO and $CO_2$ signals were only monitored for this experiment to verify carbon deposition. Water was separated through a −12 °C Peltier cold trap on the ICCS. However, the wide peak that appeared at about 450 °C was identified to be $CO_2$, as shown by the mass spectrum (Figure 10).

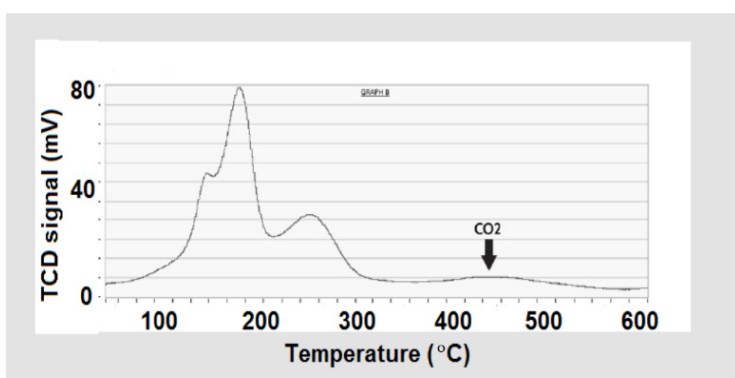

**Figure 9.** TPO profile on the spent catalyst.

For better interpretation of the results especially on the TPO profile of the spent catalyst, a fresh sample of the catalyst was reduced through TPR up to 650 °C (Figure 11). This analysis was immediately followed by a TPO analysis (Figure 12). The latest TPO profile mainly showed two different peaks. One peak is at about 150 °C which could be related to some consumption of $O_2$ as shown by the mass spectrum (Figure 13), and another peak is at about 450 °C which was not identified by the mass spectrum and could be related to the oxidation of some remaining salt traces that were used for the preparation of the catalyst. When this latter profile is compared with the TPO profile shown at Figure 9 (spent catalyst), it clearly identifies different components, if any, and that were retained or

produced during the reaction. When the mass spectrum of the fresh catalyst (Figure 13) is compared to the mass spectrum of the spent catalyst (Figure 10), it clearly shows that the spent catalyst may have carbon deposition resulting from the Bosch reaction at high temperatures. The mass spectrum of the fresh catalyst did not show the same pattern as illustrated by the TPO profile (Figure 12), which suggests the presence of non-identified ions from the fresh catalyst. It can be concluded from this last result that if the reaction time would have been extended, the catalyst would have been slowly deactivated as carbon would keep building on the surface of the catalyst blocking the entrance to the active area within the catalyst pores, and thus reducing diffusion and deactivating the catalyst.

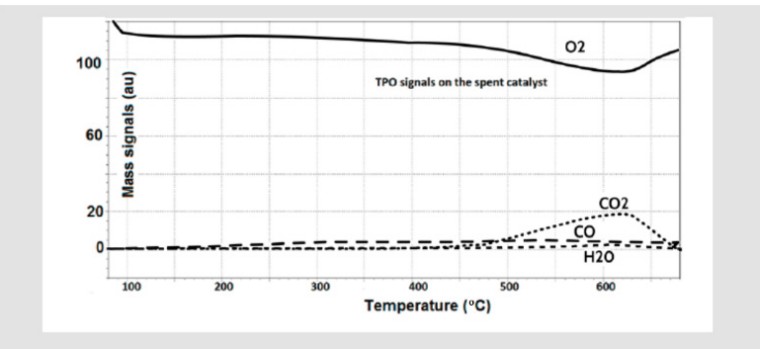

**Figure 10.** TPO spectrum on the spent catalyst.

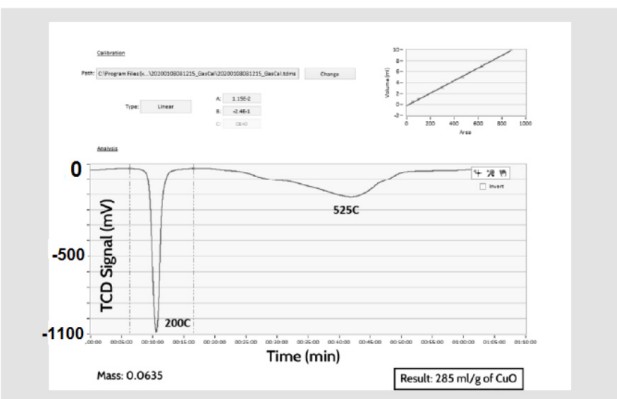

**Figure 11.** TPR profile on the fresh catalyst.

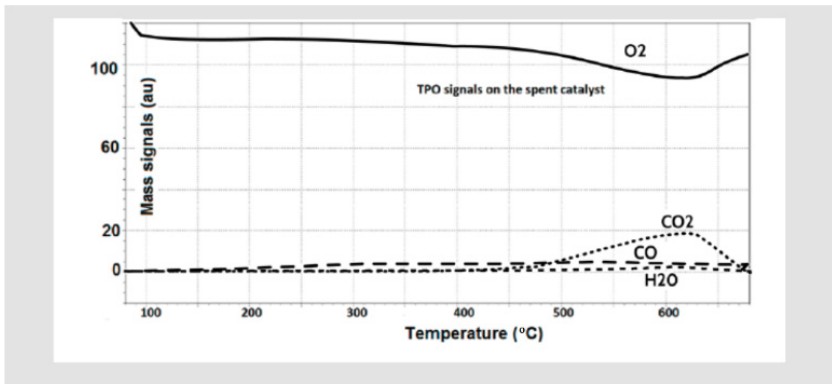

**Figure 12.** TPO profile on the fresh catalyst.

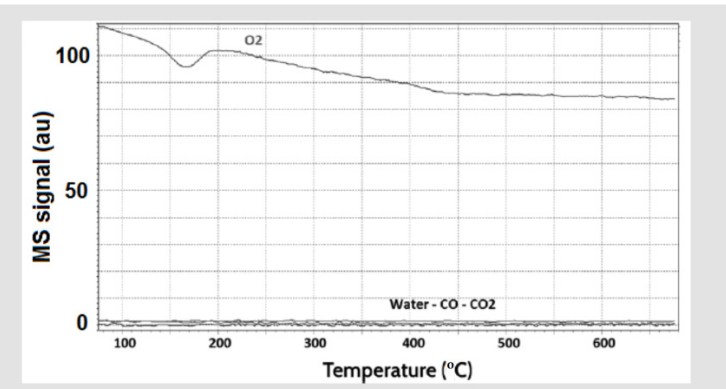

**Figure 13.** TPO spectrum on the fresh catalyst.

### 5. Conclusions

The combination of these three instruments becomes a very powerful tool for researchers, especially for those who work in the catalysis field. The results presented here demonstrated that the ICCS can perform a complete in situ characterization of the catalysts before and after the reaction. The TPR analysis not only ensured a complete reduction of the catalyst prior to the reaction, but also revealed several important pieces of information about the catalyst, including: Quantity of the oxides present on the catalyst, reduction temperature that is related to the nature of the active sites and their interaction with the support; estimation of the homogeneity/heterogeneity of active particles indicated by the width of the produced peaks, etc. However, the characterization of the catalyst after reaction is also a mandatory task that often needs to be performed. The tests after reaction, such as the TPO demonstrated here, will help reveal the causes of the catalyst deactivation. Typically, this occurs either due to sintering of active particles and loss of active area for reaction, or by carbon formation via the Bosch reaction at elevated temperatures that block the pores and reduce the diffusion of the reactants to the inner pores where the active species are vastly present.

Online monitoring by the mass spectrometer of the reaction's products permits the users to follow the different steps of the reaction as the temperature is slowly increased. This process not only reveals the optimal reaction conditions, but also actively monitors the possible deactivation of the catalyst under serious circumstances of pressure and temperature. Thus, it permits the operator, in cases when it is needed, to adjust the reaction conditions before the catalyst is completely deactivated. The obtained results shown by the mass spectrometer demonstrated that the Sabatier reaction requires a pressure of at least 30 bar and high temperature to produce the desired product. These results showed no production of $CH_4$ at atmospheric pressure and only carbon monoxide and water were produced as co-products. It was also observed by the TPO analysis that carbon was formed at high temperatures that can deactivate the catalyst with a longer reaction time. Only $CO_2$ was produced for this experiment, which could be related to the fact that copper would possibly be a good catalyst to oxidize CO into $CO_2$.

The different peaks that were not identified by either TPR or TPO could be due to the presence of some remaining traces from the salts used to prepare the catalyst.

Further characterization techniques would have been also implemented for a wider and more comprehensive study of this catalyst. Pulse of $N_2O$ for example, would have revealed the dispersion of the active species and the possibility of sintering at high temperature that is required by the reaction.

It can be concluded from this partial study that the combination of a mass spectrometer and the ICCS instrument, when connected to the Micromeritics FR micro-reactors series, become a very powerful and useful tool to perform in situ characterization and test the catalyst before and after deactivation, which is the mandatory information in catalysis that is required for the study of any reaction.

Note: The Sabatier reaction studied in this note was only for illustration purposes. It was not intended to find the optimum condition and/or the best catalyst, but to demonstrate the importance and efficacy of the use of these instruments connected together and to render them a ONE powerful tool for researchers in catalysis.

**Author Contributions:** Study conception, design, and supervision: S.Y. and J.K.; data collection: S.Y., U.P.K. and H.N.; analysis and interpretation of results: S.Y., U.P.K., H.N. and J.K.; draft manuscript preparation: S.Y.; revision of the manuscript: S.Y., U.P.K., H.N. and J.K. All authors have read and agreed to the published version of the manuscript.

**Funding:** This research received no external funding.

**Data Availability Statement:** The data presented in this study are available in article.

**Acknowledgments:** The authors acknowledge Avery Spalding, Marketing department, Micromeritics Instrument Corporation.

**Conflicts of Interest:** The authors declare no conflict of interest.

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
