# Peer review of "Exploiting In-Situ Characterization for a Sabatier Reaction to Reveal Catalytic Details"

_chemistry, doi:10.3390/chemistry3040084_

Round 1

Reviewer 1 Report

This is a very interesting piece of work in which, in a rather clear and didactical manner, the authors teach the reader how to use several techniques aiming at describing the catalytic phenomenon. The use of a classic reaction (the Sabatier reduction of CO2 in the presence of hydrogen) as well as a traditional catalyst (CuO/alumina supported commercial catalyst) is a clever way of demonstrating the potential of combined techniques to determine the mechanism of reaction and the catalyst deactivation.  As previously mentioned, the didactical merit of the article is undisputable; hence, it deserves publication.

The article is well-written. Nevertheless, it requires minor revision. Authors are kindly requested to address the following points.

1 – The excessive use of commas. For instance, in line 42: “The mass spectrometer, is connected to the exhaust of the flow reactor for online detection and quantification of the reaction products.” The comma after “spectrometer” must be erased.

In line 134: “Figure 5 represents signals for both, reactants (CO2 + H2) at 10 bar of pressure.” The comma after “both” must be erased.

A revision of the text must be performed. For instance, in line 188: “The TPO profile shows several peak …”. Should be “peaks”.

In line 201:” When this latest profile is compared…” should be: “When the latter profile is compared…”

In line 1165:” However, It was observed that larger amount of …”

In line 153 “All studied reaction produced water” reactions

ter was produced in this case,

2 – Authors say: “1.5 g of 13% CuO alumina supported commercial catalyst was used for all of the above mentioned 60 reactions”. It must be borne in mind that Ni supported catalysts are the traditional methanation catalysts. Therefore, authors are requested to add a comment on that.

3 – Authors say: “Comparisons with the TPO profile done on the fresh material after reduction (figure 12) would indicate if 185 it there was any cracking of CO2 at high temperature. Cracking would cause carbon deposition on 186 the surface of the catalyst and block access to the active sites for the reaction.”

The term “cracking” is misused here. Carbon (graphite) is formed via the traditional Bosch reaction (CO2(g) + 2 H2(g) → C(s) + 2 H2O(g)), which is a two-step consecutive reaction.

The first reaction, the reverse water gas shift reaction, is a fast one:

CO2 + H2 → CO + H2O

The second reaction is the rate determining step:

CO + H2 → C + H2O

Authors are kindly requested to correct the term “cracking” and to add a comment on the Bosch reaction.

Author Response

Please see attached document for reponse.

Reviewer 2 Report

This paper presents some interesting clues concerning the interest of the combination of different equipments (MS, micro-GC, flow reactor and an in situ characterization system) for the characerization of catalysts in situ, exemplified in a CuO/Al2O3 catalyst for the Sabatier reaction

The paper is publishable after revision according to the following suggestions:

1) Experimental section:

Co2 hidrogenation to methane reaction (CO2 + 4H2 = CH4 + 2H2O) is not shown. Please add.

The name of the reactions depicted must be added

2) The supplier of the commercial catalyst used must be given.

3) The conditions of the TPR and TPO experiments must be given (flow, H2 concentration etc.=

4) what m/z signals are used to follow the CO, CO2 and CH4?

5) At which temperature is obtained the data shown in figure 8?  If data are obtained at different temperature, please explain

6) All TPO and TPR figures must be have the same X axis in order to a best comparison

7)  Please, revise the references format of the journal.

8) Author contribution chapter is absent

Author Response

Please see attached response.
